# R-learning in actor-critic model offers a biologically relevant mechanism for sequential decision-making

**Sergey Shuvaev**[1]*, **Sarah Starosta**[1,2]*, **Duda Kvitsiani**[3], **Adam Kepecs**[1,2], **Alexei Koulakov**[1]
[1]Cold Spring Harbor Laboratory, [2]Washington University School of Medicine in St. Louis,
[3]DANDRITE, Aarhus University | sshuvaev@cshl.edu, sarah.starosta@wustl.edu,
kvitsi@dandrite.au.dk, akepecs@wustl.edu, koulakov@cshl.edu

## Abstract

When should you continue with your ongoing plans and when should you instead decide to pursue better opportunities? We show in theory and experiment that such stay-or-leave decisions are consistent with deep R-learning both behaviorally and neuronally. Our results suggest that real-world agents leave depleting resources when their reward rate falls below its exponential average, which, we argue, is a Bayes optimal rule in dynamic natural environments. Our work links reinforcement learning, the marginal value theorem and Bayesian inference approaches to offer a learning algorithm and a decision rule for making sequential stay-or-leave choices.

## 1 Introduction

In everyday life we repeatedly face sequential stay-or-leave decisions. These decisions include time investment, employment, entertainment and other choices in settings where rewards decrease over time. For example, should we continue watching a TV series although each new season is not as good as the previous one? The ways how real-world agents decide whether to commit to their current option or to search for a better one are studied in this paper.

The puzzle of real-world sequential stay-or-leave decisions has started to get attention in behavioral research only very recently (1). A few studies have explored sequential stay-or-leave decisions in humans, or rodents – the model organism used to access neuronal activity at high resolution. In both cases, decision patterns were collected in *foraging tasks* – the experimental settings where subjects decide when to leave depleting resources (2). In human studies, foraging environments were usually virtual with money used as a proxy for the reward (2). In animal tasks, reward options were represented by multiple sources of primary rewards (e.g. food, water), decreasing in size or probability over time to model natural resource depletion (2; 3).

In the aforementioned experiments, the average-case foraging decisions were consistent with the Marginal Value Theorem (MVT) (2; 4) – a theoretical framework suggesting that an agent should abandon depleting resources when the next expected reward falls below the average reward in the environment discounted by the travel-related costs (Appendix A1). This approach, however, assumes that the agents have the full knowledge of the environments, which may not be possible in real-world settings. Alternative frameworks for the models of real-world decision-making allow the agents to update their estimates of environmental variables. These frameworks, so far not often applied in a context of foraging, include model-based reinforcement learning (RL) (1) and Bayesian inference (5) – both assuming that an agent has a *model* of the environment where only a small number of relevant variables are updated with the changes in environmental states (2).

The available theoretical frameworks for sequential stay-or-leave decision-making explained some important aspects of these decisions, such as the average-case decision rules and how they may be

---

updated with changes in the environment (2). Yet, the behaviors of real-world agents may be more complex than those predicted by the existing theory. For example, both human and animals are likely to generalize experiences across different environments, suggesting that they learn meta-rules to craft their policies (6). Do theoretical average-case rules apply to real-world individual choices? Are the existing models sufficient to produce optimal decision patterns in diverse natural environments? How are the optimal decision rules learned by human and animals (1)? Here, we build a new computational model of sequential stay-or-leave decisions to address these questions.

Computational models of reward-driven behaviors are typically studied in the context of reinforcement learning (RL) (1; 2; 7). RL is the area of machine learning and artificial intelligence that deals with navigating in environments to maximize future rewards. RL models rely on a scalar signal reflecting how well / poorly an agent performs on each iteration (8). This signal serves to update a *value function* (the discounted cumulative sum of expected future rewards). The updates aim to minimize the difference between the expected and actual reward (time difference (TD) error). As the future rewards may depend on the current state and choice of action, the models select the actions to maximize the value function. Famous for successes in a wide range of artificial intelligence tasks (9; 10), RL has gained popularity in neuroscience due to its biological relevance, as described below.

Beyond successes in accounting for adaptive choices in human and animals (11), RL has helped understanding neuronal responses observed in reward-guided behaviors (11; 12). For example, activities of dopaminergic neurons in the ventral tegmental area (VTA) quantitatively reflect computation of the reward prediction error (RPE): they respond to the unexpected rewards with elevated firing rate, and respond to the reward omissions with decreased firing (13; 14). The RPE is the difference between the actual and expected rewards which numerically matches the TD error in RL. Therefore it has been proposed that the VTA dopaminergic neurons may implement RL in the brain (15). Such explanatory power made RL a widely adapted framework for interpreting experimental data and designing experiments when reward-seeking behaviors are involved (11).

One RL approach, particularly suitable for biologically relevant models of decision making, is the actor-critic framework (8). Unlike classical RL where the actions are selected to maximize the value function, actor-critic models compute the likelihoods to take an action in the "actor" part, separately from the "critic" part computing the state values. Once training is finished, the actions yielding the highest future rewards are selected probabilistically. Stochastic policy formulation, stability and continuous control made actor-critic models a well-suited tool for complex sequential decision tasks (16) and a reasonable choice for building models of real-world agents. Signatures of actor-critic algorithms have been noted in the brain, specifically at cortico-striatal pathways (17; 18).

In our work, we used the actor-critic framework to propose a mechanism that real-world agents may use to learn sequential stay-or-leave decisions. To explore the choice patterns and their neural correlates, we designed a foraging task for mice and performed recordings of dopamine neuron activity in the VTA. We compared animals' behaviors under various manipulations to a range of models and show that individual decisions rely on sequences of past rewards in a way consistent with predictions of R-learning – an RL algorithm for optimizing an average reward, weighing far-term and near-term reward the same (19; 8). In addition, we demonstrate that neuronal activity recorded in the VTA correlates with the TD error in R-learning. To establish that our results are not specific to particular choices of parameters, we provide analytical predictions for each model in experimental settings. This way, we use interpretable deep RL models to propose that learning of sequential decisions in real-world agents is consistent with R-learning, both on behavioral and neuronal levels.

## 2 Results

### 2.1 Sequential foraging decisions reveal stay-or-leave choice modulation in mice

The goal of this work was to identify the mechanism how real-world agents learn to make sequential stay-or-leave decisions in the context of depleting resources. To pursue this goal, we developed *foraging tasks* in which animals navigated between multiple sources of depleting rewards.

Water-deprived mice were allowed to run back and forth between two ports offering water reward (Figure 1A). Upon entering a port, animals received a drop of water, representing a harvest from a newly discovered resource. After consuming a reward, an animal had two choices. It could stay at the same port and get a reward constituting $\xi = 0.8$ of the previous one; this way the yield of the port was

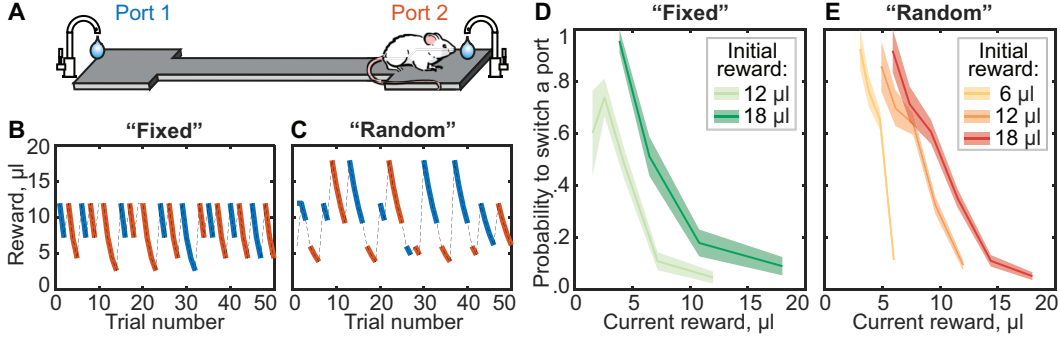

Figure 1: Foraging task for mice to study real-world sequential decisions. (A) Task scheme. (B,C) Reward patterns in the task. (D,E) Statistics of mouse behavior in the task.

decreasing exponentially (Figure 1B). Alternatively, the animal could go to another port (representing a new resource) and get a full-sized water reward. Every subsequent reward could be consumed no earlier than in 6 seconds after the previous one; travel time between the ports was relatively small (1 second). When an animal eventually returned to the first port, it received a full-sized reward. This way, we used two water ports to model depleting resources in natural environments.

Should a mouse stay at the same port for too long, it would eventually be getting only small rewards. Conversely, should a mouse run between ports after every trial, it would maximize the consumed water, but would also expend a high amount of energy to do so. The optimal strategy likely involves staying at each port for a few trials and then leaving for a new port. Such strategy may provide clues into the animals' decision rules and the ways animals may learn these rules.

**Animal task 1** ("fixed initial rewards"). First, we tested whether animals' behavior follows the predictions of the MVT and is in line with previous results showing that real-world agents tend to leave depleting reward sources at higher thresholds in richer environments (2; 3). In one session, we offered the animals the initial rewards of $12\mu l$ of water (Figure 1B). In another session, the initial rewards were increased to $18\mu l$ of water. To reveal the statistics of mouse behavior we extracted the probabilities to leave a port as a function of the preceding reward amount (Figure 1D). We show that animals tended to leave a port at a higher reward threshold when the initial reward was larger (Figure 1D). This result is consistent with prior experimental observations and theoretical conclusions attributing the leaving reward threshold to the average reward in the environment.

**Animal task 2** ("random initial rewards"). To gain insights into the animals' individual stay-or-leave decisions, we designed a novel reward schedule. Mice received the initial rewards of $6/12/18\mu l$ of water drawn randomly *after every switch of ports* to model uncertainty in natural resources and to decorrelate the reward sequences for analysis (Figure 1C). We observed that larger random rewards immediately after entering a port (e.g. $18\mu l$) resulted in leaving the port at a higher reward threshold (Figure 1E). This result implies that the animals, when offered larger reward, tended to leave the beneficial option. Below, we present computational models to explain this counterintuitive behavior.

## 2.2   Actor-critic models offer a potential learning mechanism used in stay-or-leave decisions

To gain insights into the learning mechanisms used by real-world agents towards depleting rewards, we built reinforcement learning (RL) models of our foraging tasks in which stay-or-leave choice patterns of mice were observed. We considered a range of learning approaches potentially used by the animals in the tasks. To single out a biologically relevant mechanism, we compared the predictions of our RL models – backed by the analytical derivations – to the choice patterns of the animals.

We built the models of our foraging tasks using the deep actor-critic framework (8) (see Methods) as follows (Figure 2A). In the models, an agent (representing mice in the animal tasks) has to decide when to leave a port yielding an exponentially depleting reward (representing water drops in the animal tasks). As an input, our models received an agent's state $\vec{s}_t$, defined as a vector of 20 latest *effective* rewards and the latest action $\tilde{r}_{t-1}, ..., \tilde{r}_{t-20}, a_{t-1}$. The effective rewards $\tilde{r}_t$ were equal to physical rewards $r_t$ (representing water drops in the tasks) discounted after the "switch" actions by the port switching costs $c_{sw}$: $\tilde{r}_t = r_t - c_{sw}$ if $a_t = $ "switch", and $\tilde{r}_t = r_t$ if $a_t = $ "stay". The input sequence of the rewards encompassed any rewards received in the environment regardless of the source (i.e. both ports and switching costs). Providing agents with information about the past rewards

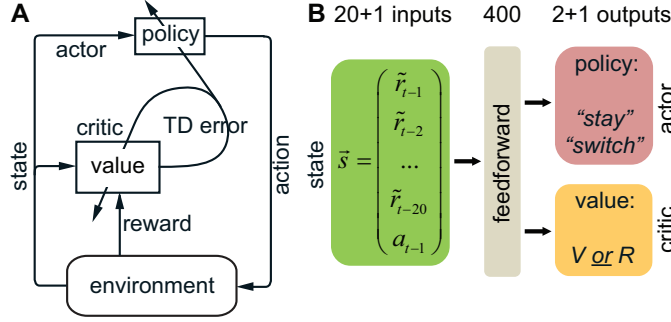

Figure 2: Actor-critic model of our foraging tasks. (A) Scheme of the actor-critic model (adapted from (8)). (B) Feedforward neural network implementing this scheme.

enabled them to learn various behaviors following the same latest reward – potentially matching the observations in the "random initial rewards" animal task (Figure 1C). The information about the latest action allows the models to distinguish the expected rewards from violations of the reward schedule ("surprises"). The state inputs were propagated through a feedforward neural network (Figure 2B) consisting of a hidden layer (sigmoid units) and a linear output layer. Neural network allowed the models to generalize their predictions over the 21-dimensional input state space – the task intractable by the means of conventional tabular RL. The first two outputs of the network, the "actor" part, represented log likelihoods of two available actions: to "stay" at the same rewarding port, or to "switch" to another port. On every iteration, an action $a_t$ was selected in accordance with these log likelihoods. In case of the "stay" action, in the next trial, the agent received $\xi = 0.8$ of the previous physical reward: $\tilde{r}_{t+1} = r_{t+1} = \xi r_t$. In case of the "switch" action, the agent paid the switching cost and received the full-size reward: $\tilde{r}_{t+1} = r_{t+1} - c_{sw}$ where $r_{t+1} \sim \{r^{init}\}$. We added Gaussian noise to rewards to represent perceptual noise in mice (see Methods).

The 3rd output of the network (the "critic" part) represented the value of current state of the agent. In this work, we considered two possible state value functions within our actor-critic model. In one set of the experiments ("V-learning"), the network's 3rd output computed an estimate of the V-function, i.e. the cumulative sum of discounted future rewards (and costs), standard for actor-critic models (8):

$$V(\vec{s}_t) = \mathbb{E}\left[\sum_{\tau=0}^{\infty} \gamma^\tau \tilde{r}_{t+\tau}\right] \text{ s.t. } \vec{s}_t = [\tilde{r}_{t-1}, ..., \tilde{r}_{t-20}, a_{t-1}]; \tilde{r}_t = r_t - \begin{bmatrix} c_{sw}, & \text{if } a_t = \text{"switch"} \\ 0, & \text{if } a_t = \text{"stay"} \end{bmatrix} \quad (1)$$

Here $\gamma$ is a temporal discount factor devaluing the future rewards. In another set of the experiments ("R-learning"), the network's 3rd output computed an estimate of the R-function, i.e. the cumulative difference between the next expected and exponentially averaged past rewards (and costs) (8; 19):

$$R(\vec{s}_t) = \mathbb{E}\left[\sum_{\tau=0}^{\infty} \gamma^\tau (\tilde{r}_{t+\tau} - \tilde{r}_{t+\tau-1}^{avg})\right] \text{ s.t. } \tilde{r}_t^{avg} = (1-\kappa)\sum_{\tau=0}^{\infty} \kappa^\tau (\tilde{r}_{t-\tau}) \quad (2)$$

Here $\kappa$ is the exponential averaging constant. Although temporal discount $\gamma \neq 1$ is seldom used in R-learning, we kept it here for completeness. Both choices of the value function are biologically realistic (8; 19). To train the networks, we used on-policy model-free time difference (TD) learning [9]. In case of the V-learning, we used the Bellman equation to compute the TD error $\delta$ reflecting the discrepancy between the expected and received rewards (and costs) (8):

$$\delta_V(t) = -V(\vec{s}_t) + \tilde{r}_t + \gamma V(\vec{s}_{t+1}) \quad (3)$$

In case of the R-learning, the TD error included an additional variable – the exponential average of past rewards $\tilde{r}_t^{avg}$ (Equation (2)) which was computed explicitly using the inputs throughout training.

$$\delta_R(t) = -R(\vec{s}_t) + \tilde{r}_t - \tilde{r}_{t-1}^{avg} + \gamma R(\vec{s}_{t+1}) \quad (4)$$

We backpropagated the TD errors through the outputs representing the state value and the latest action (to update state value and policy separately using semi-gradient descent (8); see Methods). We trained and evaluated the models in two types of reward schedules, as follows.

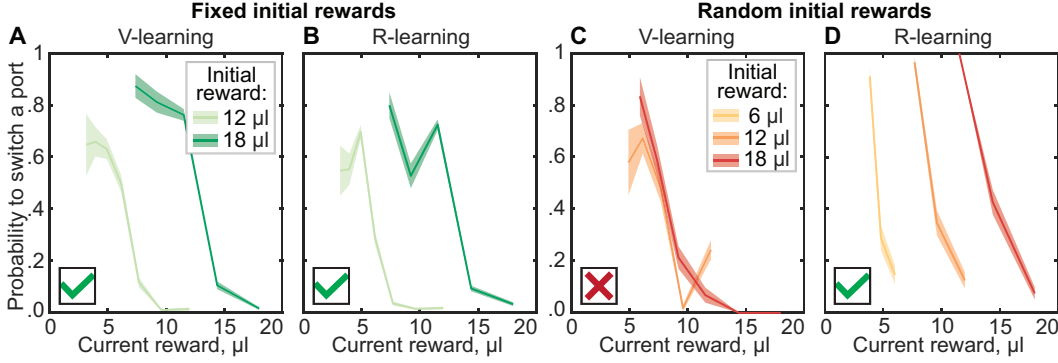

Figure 3: Predictions of V- and R-learning models in conventional and novel task. (A,B) Both models explain observations in conventional task. (C,D) Only R-model explains observations in the novel task.

**Experiment 1** ("fixed initial rewards"). First, we tested whether our models reproduce the behaviors of mice towards two ports with a fixed initial reward. Our data (Figure 1D) suggests in line with prior results (2) that in sessions with larger initial rewards animals tend to leave the ports at a higher reward threshold. Here, we trained V- and R-learning models in the actor-critic framework using the initial rewards of 12 or $18\mu l$ (only one value was used for the entire experiment, e.g. always $12\mu l$). As observed experimentally, in both V- and R-learning models the agents tended to leave the ports at a higher reward threshold in sessions with higher initial rewards (Figure 3A,B). This way, the agents' policy in both models reproduced the animals' behavior in the "fixed initial rewards" task.

To explain these results, we derived analytical predictions for optimal stay-or-leave policies in V- and R-learning models. We based our theory on the Marginal Value Theorem (MVT) (2; 4) – an established framework for optimal foraging behavior:

**Theorem 1** (the marginal value theorem, MVT). *An agent maximizes its average reward intake if it leaves depleting reward sources when the next expected reward $\tilde{r}_{n+1}$ falls below the average reward in the environment $\langle \tilde{r}_n \rangle$ computed over rewards $\{r\}$ and travel costs $\{c_{sw}\}$.*

$$\tilde{r}_{n+1} = \langle \tilde{r}_n \rangle \tag{5}$$

The proof for the MVT (4) can be found in (Appendix A1). We used Theorem 1 to show that:

**Theorem 2** (the V-learning optimum). *In the limit of low temporal discount $\gamma \to 1$, the optimal policy for a V-learning agent navigating between depleting reward sources is defined by the MVT.*

Indeed, the objective of the V-learning is to maximize the cumulative future discounted reward (8). In the limit of low temporal discount, V-learning maximizes the total future reward, or the reward rate multiplied by total time, thus aiming the optimum predicted by the MVT (see Appendix A2).

We used Theorem 2 to derive the optimal policy for a V-learning agent (Appendix A2). The number of trials $n$ spent in a particular port is related to the the initial reward $r_0$ by the following equation:

$$r_0 = c_{sw} \left( \frac{1 - \xi^n}{1 - \xi} - n\xi^n \right)^{-1}, \tag{6}$$

where $\xi = 0.8$ is the reward depletion rate. Although this equation is not invertible analytically, the number of trials $n$ in a port as a function of $r_0$ can be found as a numerical inverse of this equation.

The threshold $r_{sw}$ at which a V-learning agent leaves a port can be written as $r_{sw} = r_0 \xi^{n(r_0)}$ which is an increasing function of $r_0$ for small values of $n$ observed in animal tasks. It is thus optimal for V-learning agents to leave ports at a higher reward threshold in sessions with larger restart values $r_0$.

We then applied the MVT to R-values – the differences between the next expected and exponentially averaged past rewards – to derive the optimal decision rule for the R-learning (Appendix A3):

**Theorem 3** (the R-learning optimum). *The optimal policy for an R-learning agent navigating between sources of depleting reward in the limit of low temporal discount $\gamma \to 1$ is to leave a source of depleting reward when the next expected reward $\tilde{r}_{n+1}$ at the source falls below an exponential average $\tilde{r}_n^{avg}$ of the previous rewards and travel costs – the decision rule we called the **Leaky MVT**:*

$$\tilde{r}_{n+1} = \tilde{r}_n^{avg} \tag{7}$$

We used Theorem 3 to derive the optimal policy for an R-learning agent in the "fixed initial rewards" experiment (Appendix A3). In the limit of $\kappa \to 1$, it matches the Equation (6) for the V-learning.

$$r_0 = \kappa^{n-1} c_{sw} \left( \frac{\kappa^n - \xi^n}{\kappa - \xi} - \frac{1 - \kappa^n}{1 - \kappa} \xi^n \right)^{-1} \qquad (8)$$

Similarly to the V-learning model, R-learning agents tend to leave ports at a higher reward threshold in sessions with higher restart values. Our theory predicts that both V- and R-learning are consistent with animal choices in the "fixed initial reward" task and similar experimental results reported previously.

**Experiment 2** ("random initial rewards"). To distinguish between the V- and R-learning models, we applied them to our novel reward schedule where initial rewards in each port were drawn randomly from ($6/12/18\mu l$) introducing an unexpected, random initial reward after every "switch". In this task, counterintuitively, mice tended to leave reward ports shortly after larger random initial rewards. This behavior seemingly contradicted the MVT in which the optimal leaving reward threshold is defined by the average reward rate, constant for each given environment. In agreement with this intuition, since we have shown that the V-learning optimum is predicted by the MVT (Theorem 2), our V-model agents were leaving the ports at a fixed reward threshold regardless of the initial random reward (Figure 3C). In contrast, the R-model reproduced the observed mouse behavior (Methods, Figure 3D). To understand this behavior of the R-model, we used Theorem 3 to describe the switching reward thresholds $r_{sw}$ for each latest initial reward value $r_0$ (Appendix A4; $\langle \cdot \rangle$ denotes the average values):

$$\text{sgn}(r_{sw} - \langle r_{sw} \rangle) = \text{sgn}\left( (\log r_0 - \log\langle r_0 \rangle)(1 - \kappa) \right) \qquad (9)$$

Equation (9) suggests that R-learning agents facing "random initial rewards" leave the ports at higher reward thresholds after the larger random initial rewards. This result, observed in the animal data and R-learning experiments, holds if $\kappa > \xi$, i.e. forgetting is slower than reward decay. In case of V-learning, $\kappa = 1$ results in leaving ports at a same threshold regardless the initial reward value.

Additionally, to compare the MVT and the Leaky MVT quantitatively, we performed the parameter fitting for both models using behavior patterns of 7 mice observed in the "random initial rewards" task. We minimized the negative log likelihood computed over the models' predictions (Appendix A5) w.r.t the parameters of the models. This procedure is similar to logistic regression, except the Leaky MVT parameters are not independent, necessitating use of gradient descent instead of deriving the exact solution. We found that, according to the Leaky MVT, mice averaged the past reward with the exponential averaging constant $\kappa = 0.87 \pm 0.05$; perceived the switching cost as $c_{sw} = 4.0 \pm 1.2\mu l$, and exhibited the decision noise of $\alpha = 0.38 \pm 0.17$. We further applied the Akaike information criterion (AIC) to the fitted data and observed that the Leaky MVT decision rule explains the data better than the MVT ($\Delta AIC = -67 \pm 40$), in agreement with our previous qualitative conclusions.

Based on these results, we conclude that mouse sequential stay-or-leave behavior in foraging tasks may be acquired via R-learning. We propose that real-world agents compare the next expected reward to an exponential average of past rewards – the decision rule we named the Leaky MVT for similarity with the conclusions of the Marginal Value Theorem (MVT). Below, we show data supporting the idea that R-learning takes place in the brain. We further discuss the reasons why real-world agents may use R-learning instead of V-learning known to maximize cumulative future reward.

## 2.3 Mouse dopamine responses are consistent with R-learning

Growing evidence suggests that, in mammalian brain, reward-guided behaviors are learned in ways consistent with RL (13; 14; 15). There are indications that such learning takes place at cortico-striatal synapses consistent with the actor-critic framework: the policy-related "actor" part is typically attributed to striatum, whereas the value-related "critic" part likely includes the ventral tegmental area (VTA) (17; 18). Dopaminergic neurons in the VTA modulate their activities consistently with the error in reward prediction, or TD error (15). To see whether such activity in the brain is compatible with the TD errors in our models, we performed an additional experiment as follows.

**Animal task 3** ("surprising rewards"). Mice were allowed to navigate between two ports which offered the fixed initial value of water reward ($12\mu l$). The reward was depleting while an animal stayed in a port. To induce learning in animals, we provided them (in 3% of trials) with surprising large / small rewards ($\pm 50\%$ to the usual reward; we assume that learning only takes place when

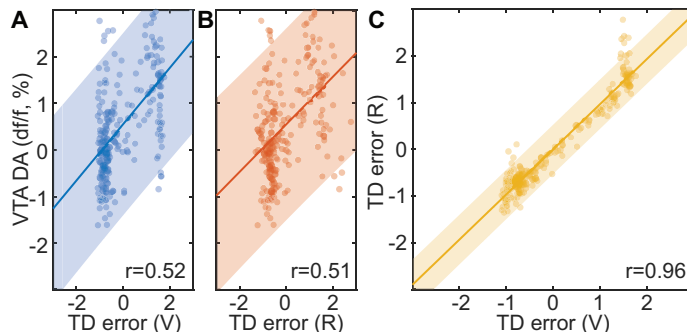

Figure 4: VTA dopamine activations (VTA DA) and TD errors. Correlations between: (A,B) VTA DA and z-scored TD errors of V/R-learning; (C) Z-scored TD errors of V- and R-learning models.

the reward amount is unexpected). We used fiber-photometry to record the activity of dopaminergic neurons in the VTA throughout the task (cre-dependent expression of GCaMP6f in DAT-Cre mice).

**Experiment 3** ("surprising rewards"). We evaluated our pretrained actor-critic models ("fixed initial rewards" of $12\mu$l) on reward sequences obtained by mice in the "surprising reward" animal task. For the surprising rewards, we collected the TD errors in both V- and R-learning. We computed the correlations between these TD errors and VTA dopamine neuron activations (VTA DA), then evaluated them at their peak values within trials (Figure 4A,B). Correlations between VTA DA and TD errors were similar for V- and R-learning ($r = 0.52$ and $r = 0.51$). TD errors in the models were also highly correlated with each other ($r = 0.96$; Figure 4C). Below, we explain this observation.

In the V-learning, the TD error can be viewed as a difference between the actual and the expected rewards. In the R-learning, the TD error similarly equals to the actual minus the expected rewards both discounted by the average past rewards. The average past rewards, computed over the past trials and unaffected by the current reward, cancel out. Thus, the VTA dopamine neuron signal can be equally well explained by the V- and R-learning TD errors, fully correlated with each other. This observation indicates that the VTA may be involved in R-learning; it may provide the TD error signal consistent with the R-learning and implementing the Leaky MVT decision rule in the brain.

Overall, our results suggest that real-world agents are likely to use R-learning in sequential stay-or-leave decisions. This conclusion is based on our behavioral and neuronal data from a novel foraging task, and emerged in our deep actor-critic model (see Methods). Our results are consisted with prior experimental data (see above). In the section below, we go through our results and discuss additional considerations addressing biological relevance of the identified learning mechanism (R-learning).

## 3 Discussion

In real-world conditions, we often face sequential stay-or-leave decisions about whether to engage with the current option, or to search for a better one. To study how such decisions may be learned, we confronted mice with the choice when to abandon depleting resources. We show that individual stay-or-leave decisions – and dopaminergic neuronal firing in the VTA of the animals – are consistent with the R-learning, an RL paradigm maximizing the difference between the next expected and exponentially averaged rewards, aiming to behave better than on the average. We further derived the Leaky MVT – a novel decision rule based on exponential filtering of past rewards. We show that this rule is implemented by R-learning (Appendix A3) and accounts for animals' behavior in our tasks. Below we discuss how these findings connect to decision-making and learning in real-world agents.

### 3.1 R-learning explains both known and newly discovered mouse behaviors

A limited number of studies have investigated stay-or-leave decisions in a foraging context with decreasing rewards. Reward schedules involved blocks of trial with fixed parameters (2) exposing animals to repeated conditions and enabling them to exercise similar choices. The resulting decision patterns were autocorrelated (20) obstructing causal inference for individual choices and favoring comparisons between statistics of different blocks of trials with constant initial rewards (2). To study individual stay-or-leave choices of real-world agents, we randomized the rewards in our task by drawing random initial rewards every time an agent switches a port. Our design aimed to: 1) reduce

autocorrelation in reward sequences to enable analyses as logistic regression, and 2) disentangle the impact of the initial and average reward. The resulting decision patterns that we observed in mice were diverse and, as such, led to new insights into the animals' behavior.

We observed that the animals, when offered one of the higher random initial rewards, surprisingly tended to leave the current port at a higher reward threshold. In other words, for the same value of current reward, the stay-or-leave decisions depended on what happened a few trials ago. To enable our models reproduce this observation, we supplied them with sequences of past rewards serving as a state, in contrast with scalar states used previously (2). With our models we found that paradoxical escape from a beneficial option could be explained with the R-learning but not with the V-learning algorithm, thus offering a new decision model potentially used by real-world agents. Our R-model generalizes predictions over various reward schedules. In particular, it explains the overharvesting, i.e. real-life agents staying at reward ports longer (2) than predicted by the MVT. This is because the exponentially averaged reward drops in parallel with the current reward – after which they match at a lower threshold than predicted by the MVT. We further set up additional manipulations of task parameters (inter-trial delays, reward depletion rates, travel paths) and considered additional RL models (port-value, action-value models) observing that our conclusions hold for a larger scope of mouse behaviors (data not presented here). Overall, deep R-learning is consistent with mouse sequential stay-or-leave decisions in a variety of conditions, accounting for individual choices of real-world agents and offering a biologically relevant mechanism for learning to make these decisions.

### 3.2 Model-free deep R-learning agents learn generalized decision rules in their connectivity

In RL, real-world decisions were typically studied in the context of model-based approaches (1), although model-free approaches were also attempted. The uncertainty of whether real-world agents use model-based or model-free RL was caused by the following lines of evidence. On one hand, activities of dopaminergic neurons in the VTA resemble the TD errors in model-free RL (21). On the other hand, animals are likely to maintain the history of their actions and rewards in prefrontal cortex (PFC) (22; 23) potentially maintaining the models of the environment. Further data suggested that model-free and model-based algorithms may coexist in dorsomedial and dorsolateral corticostriatal loops in the brain (24). This duality of model-based and model-free approaches has been explained recently with the use of deep RL (6). Model-free deep neural networks trained under diverse conditions formed meta-learning circuits, successfully grasping the models of novel environments and behaving optimally in those without additional learning. We build upon these results and offer a model-free deep RL algorithm (8) reproducing model-based sequential decisions of real-world agents. Our model develops policies consistent with the Leaky MVT behavior of animals in the "random initial rewards" task. Indeed, our actor-critic R-learning network can adapt to new values of initial rewards and change its stay-or-leave policy based on the current value of exponentially averaged past rewards. This adaptation occurs without any changes in the model weights, similarly to the meta-learning agents described previously (6). Our results thus extend the body of previous works that argue the model-based vs. model-free dichotomy being unnecessary (6; 25).

Deep RL has also allowed us to establish correspondence between the MVT-based and RL-based models of sequential stay-or-leave decision-making (Theorems 2, 3). We show that the optimal deep V-learning agents implement the MVT decision rule (Appendix A2), whereas the optimal deep R-learning implements the rule predicted by our newly introduced Leaky MVT (Appendix A3). This result became possible because deep RL agents can interpolate their policies over multidimensional state spaces, e.g. sequences of rewards, providing models with sufficient data to infer statistics of complex tasks (26) and thus to derive the optimal policies reported in theoretical works.

### 3.3 R-learning and optimality of Bayesian inference in dynamic real-world environments

Sequential stay-or-leave decisions of animals in the "random initial rewards" task resemble the predictions of the R-learning model (Appendix A4) and can be explained with the decision rule that we called the Leaky MVT. This rule suggests that animals, to make a stay-or-leave decision, compare the next expected reward to the exponential average of their previous rewards (Appendix A3). The Leaky MVT rule, relying on the exponential average of rewards, is different from the canonical MVT rule (Appendix A2). This discrepancy suggests that the animals in our foraging task optimize something different than the flat average rate of reward, as proposed by the MVT (Appendix A1).

Several studies used hidden Markov models (HMMs) to show the optimality of exponential filtering for maximizing reward in dynamic environments (27; 28). Similar ideas can be applied to foraging tasks once they are viewed as the evidence accumulation problems (5). To behave optimally in dynamic natural environments, the agents need to update their estimates of environmental variables. In the case of foraging, the animals need to update their estimates of average reward, as it may change over time. These updates may be performed by maintaining a leaky average of past rewards, as suggested by the Leaky MVT. On evolutionary timescales, the animals have been exposed to dynamic natural environments, for which they have finetuned their behaviors. This way, the animals may use universal meta-rules, exhibiting them in static experimental setting by extension (28). We propose that the Leaky MVT may embody such meta-rule allowing real-world agents to make optimal stay-or-leave choices in dynamic environments.

The Leaky MVT may be advantageous for computation. Once learned, the Leaky MVT decision rule enables the agents to adjust to changes in reward contiguity without need to update high-dimensional state values. It allows a simple implementation: the exponential averaging may be performed with a single recurrent neuron. Such update mechanism may not only be computationally efficient ($\mathcal{O}(n)$), but also has potential of outperforming more complex methods in time-varying stochastic environments. This is akin to (29) showing that an $\mathcal{O}(n)$ gain tuning beats the $\mathcal{O}(n^2)$ Kalman filter in dynamic environments, because the latter is only optimal if the model of the world is precise. Overall, as the Leaky MVT emerges from the R-learning (Appendix A3), we argue that the R-learning has a potential to offer an optimal strategy for sequential stay-or-leave decisions in real-world conditions. Studies of such optimality may form an interesting direction for the future research.

### 3.4 TD error in R-learning is consistent with V-learning and VTA dopamine literature

We show in theory and in experiment that dopamine signaling in the VTA correlates with the TD errors of both V- and R-learning. This is because the teaching signals of the V- and R-learning are fully correlated. The VTA dopamine responses, previously attributed to the TD errors of the V- or Q-learning (13; 15), may be therefore interpretable as the TD errors in R-learning. Despite the same teaching signal, the V- and R-learning yield different policies. This is because, the TD update is different for the V- and R- models before they converge to an optimal policy. In the case of a fully learned policy, the TD errors are close to zero both in V- and R-learning models.

The MVT and the Leaky MVT – the decision rules we prove optimal for the V- and R-learning respectively – belong to the class of reference-based valuation frameworks, known to the neuroscience community. The models in this class evaluate the rewards w.r.t their average values. Early theoretical works argued that rewards in real-life environments are often related to each other, and keeping track of the average reward in the environment may be a biologically plausible way of accounting for the reward contingency (30). The average reward was hypothesized to be encoded in tonic serotoninergic (30) or dopaminergic (31) activity. Other theoretical accounts suggested the role of the average reward in control of meta-learning parameters (32). Several studies combining theoretical and experimental results have shown that accounting for average reward in RL explains the changes in animals' response vigor (31) and predicts motivation-related neuronal activity in the ventral pallidum (33), thus connecting reference-based RL to motivation behaviorally and neuronally.

In this study, we present a direct evidence showing that the short-term reward history modulates the decisions towards the identically-valued current rewards. Local variations which we introduced in the reward schedule allowed us to observe that mouse decisions depended on ∼10 trials, introducing a significant choice variability within a single session (∼100 trials) and arguing that the animals may perform a short-term averaging of the reward. We further built a theoretical connection between the models in the field (the V-, R-learning, the MVT and the novel Leaky MVT; see Appendix A2,A3) and show that both deep V and R-models converge to decision rules where the next expected reward is compared to an average reward in the environment. Thus, both approaches can be viewed as the reference-based valuation methods, yet only the deep R-learning produces the behavioral patterns consistent with the experiments, emphasizing the role of short-term reward averaging.

Overall, our work links the marginal value theorem, reinforcement learning and Bayesian inference to offer a decision rule and learning mechanism for real-world sequential stay-or-leave decisions. Our model offers an interpretable single-trial level description of the stay-or-leave decision process, enabling further targeted investigation into the structure of decision-making circuits in the brain.

## Broader Impact

In this work, we used a new foraging task involving decorrelated reward sequences which enabled causal inference of stay-or-leave decision rules in behaving agents (as opposed to the average-case rules studied previously). We propose a biologically relevant decision rule and the corresponding learning mechanism accounting for individual decisions of real-world agents. The authors are not aware of any potential negative societal or other consequences of the new results introduced in this work. All animal procedures in this work were approved by the Cold Spring Harbor Laboratory Institutional Animal Care and Use Committee in accordance with the National Institutes of Health regulations.

## Acknowledgement

We thank Aubrey Siebels for technical assistance with conducting the animal experiments. Funding in direct support of this work: The Swartz Foundation; DFG Grant STA 1544/1-1. Additional revenues related to this work: Travel support by The Simons Center for Quantitative Biology, The Gatsby Charitable Foundation, Burroughs Wellcome Fund, Google DeepMind, and Simons Foundation.

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
