[Supplementary Material]

## Methods

To study the ways of how real-world agents may learn to make sequential decisions, we built models of the animal foraging tasks described in this paper. We used the deep actor-critic framework to define the agent navigating in a simulated environment. On every iteration, the agent received as an input the sequence of 20 latest *effective* reward values $\tilde{r}_t$ (i.e. the *physical* reward values $r_t$ discounted by the switching costs $c_{sw}$ – see below) and the latest action $a_{t-1}$ – together serving as the agent's state $\vec{s}_t$.

$$\vec{s}_t = [\tilde{r}_{t-1}, ..., \tilde{r}_{t-20}, a_{t-1}]; \tilde{r}_t = r_t - \begin{bmatrix} c_{sw}, & \text{if } a_t = \text{"switch"} \\ 0, & \text{if } a_t = \text{"stay"} \end{bmatrix} \tag{1}$$

This sequence (normalized to $\tilde{r} \to \tilde{r}/6 - 1$; $a \sim \{0 = \text{"stay"}; 1 = \text{"switch"}\}$ so that the input was roughly zero-mean and has the variance of 1) was propagated through feedforward neural network (a hidden layer of 400 sigmoid units, and a linear output layer of 3 units; initialized using the Xavier rule). The outputs of the network were interpreted as: 1) the state value, and 2-3) the log likelihoods $L_{st}$ to "stay" at the source of rewards and $L_{sw}$ to "switch" to another one. We used these log likelihoods to draw the agent's actions: $a_t \sim \text{Bernoulli}(\exp L_{sw}/(\exp L_{sw} + \exp L_{st}))$.

We communicated the selected action to the simulator of the environment, so that it could provide the agent with an appropriate reward. Whenever the "switch" action was selected, the environment re-initialized the *physical* reward $r_t$ with one of the starting values $\{r_{init}\}$; otherwise it yielded $\xi = 0.8$ of the previous reward value. We added 2 sources of zero-mean Gaussian noise to *physical* reward values $r_t$ to model perceptual noise in animals: 1) "Weber-Fechner" noise ($\sigma_1 = 0.1$) proportional to the current reward (after "switch") or to the difference between the previous and current rewards (after "stay") and 2) constant "background" noise ($\sigma_2 = 1$). Noisy reward depletion was applied to the noisy values of previous rewards (Equation 2). The *effective* reward values $\tilde{r}_t$, communicated to the agent (Equation 1), included the switching cost $c_{sw} = 18$.

$$r_t = \begin{bmatrix} r_0 + \Delta_1 \cdot r_0 + \Delta_2, & \text{if } a_t = \text{"switch"} \\ \xi r_{t-1} + \Delta_1 \cdot (1 - \xi) r_{t-1} + \Delta_2, & \text{if } a_t = \text{"stay"} \end{bmatrix} \text{ s.t. } \begin{cases} r_0 \sim \{r_{init}\} \\ \Delta_1 \sim \mathcal{N}(0, \sigma_1) \\ \Delta_2 \sim \mathcal{N}(0, \sigma_2) \end{cases} \tag{2}$$

We used these rewards to update the network's outputs as follows. First, we computed the TD error $\delta(t)$. To do so, we defined the agent's new state $\vec{s}_{t+1}$ using the newly received *effective* reward $\tilde{r}_t$ and $20 - 1 = 19$ previous rewards, then re-evaluated the outputs of the network using this new state. We used the new and the previous state values in the Bellman equation to compute the TD error. In case of the V-learning, the TD error was equal to the latest *effective* reward minus the discrepancy between previous and discounted new state values $V(\cdot)$. We used the temporal discount coefficient $\gamma = 0.9$.

$$\delta_V(t) = -V(\vec{s}_t) + \tilde{r}_t + \gamma V(\vec{s}_{t+1}) \tag{3}$$

In case of the R-learning, the *effective* reward $\tilde{r}_t$ in the Bellman equation was decreased by the value of the exponentially averaged *effective* reward $\tilde{r}_{t-1}^{avg}$, computed over the inputs throughout training with the constant of exponential averaging $\kappa = 0.9$. We refer to the state values in R-learning as $R(\cdot)$:

$$\delta_R(t) = -R(\vec{s}_t) + \tilde{r}_t - \tilde{r}_{t-1}^{avg} + \gamma R(\vec{s}_{t+1}) \text{ s.t. } \tilde{r}_{t-1}^{avg} = \frac{(1 - \kappa)}{(1 - \kappa^{20})} \sum_{\tau=0}^{19} \kappa^{\tau} (\tilde{r}_{t-1-\tau}) \tag{4}$$

The TD error was assigned as the teaching signal to the network's outputs reflecting the value function and the latest action. We backpropagated the teaching signals through the network activations corresponding to the previous state (semi-gradient update). We performed $N = 3 \cdot 10^5$ iterations per experiment with the learning rate decaying linearly from $10^{-4}$ to 0. To overcome the temporal credit assignment problem, we grew the switching cost throughout training iterations $(n)$ from $0.5c_{sw}$ to its full value: $c_{sw}(n) = c_{sw} \cdot \text{sigmoid}(10n/N)$ as the larger switching costs necessitated accounting for more future iterations to behave optimally. Training could take multiple restarts to converge.

In the scope of this work, we performed 2 types of training. In the 1st set of experiments, we trained the models with a "fixed initial reward" value (either always 12 or always $18\mu l$). In a separate 2nd set of experiments, we trained the models with the "random initial reward" values after every "switch" action ($6/12/18\mu l$). In both cases, we trained V- and R-models, 10 restarts per experiment. Each

experiment took 2.5 minutes on Intel Core i7-based laptop computer using 1 GB RAM. To evaluate the policies learned by the models, we ran each trained model 20 times, for 100 iterations each time (to match the animals' data containing the behavioral patterns of 4-7 mice in $\sim$20 sessions of $\sim$100 iterations in 3 tasks), then we plotted the average probabilities to leave a reward source as a function of the preceding reward. We only displayed the data points with 6 or more observations. The policies of each animal and of each model were evaluated separately.

For the "surprising rewards" testing, we used the pretrained model with a "fixed initial reward" value (12 $\mu$l) which we evaluated on reward sequences obtained in the animal task. In this data, in 3% of the trials the reward values were 50% larger or smaller that we expected ones. For such trials, we recorded the TD errors in our V- and R-learning models ($\delta_V$, $\delta_R$) to be compared with the dopamine signal in the VTA ($df/f$). We normalized the TD erros for display purposes (zero-mean, unit variance across all surprising trials), then computed Pearson correlations of TD-errors and the VTA DA signal for every time point within a trial. We recorded the maximum values (over time) of these correlations.

To compare the MVT and the Leaky MVT quantitatively, we performed parameter fitting for these models using behavior patterns of 7 mice in the "random initial rewards" task. We extracted all behavior patterns of the length 21 from the data (rewards and actions) and performed gradient descent on the negative log likelihood computed over the models' predictions (Appendix A5) independently for each animal. We performed $10^5$ optimization steps used the learning rate of $10^{-6}$. The initial values of the parameters for the MVT were sampled randomly from $\mathcal{N}(0,1)$; for the Leaky MVT we started with $\alpha = 1; \kappa = 0.9; c_{sw} = 5$. The result did not change for the initial values of $\alpha = 0.1...2; \kappa = 0.5...1; c_{sw} = 1...10$. We compared the fitted models using the Akaike information criterion (AIC). We further derived analytical predictions for both models in our experimental settings.

# A   Appendix

## A.1   The marginal value theorem (MVT)

**Theorem 1** (the marginal value theorem, MVT). *An agent maximizes its average reward intake if it leaves depleting reward sources when the next expected reward $\tilde{r}_{n+1}$ falls below the average reward in the environment $\langle \tilde{r}_n \rangle$ computed over rewards $\{r\}$ and travel costs $\{c_{sw}\}$.*

*Proof.* Here, we follow Eric Chanov's 1976 paper. We provide this proof here for completeness.

The goal of the agent here is to optimize the average reward intake rate $\mathfrak{R}$. The average reward intake rate depends on rewards $r_m$ received by the agent on each sequential iteration $m \in \{1...n\}$ at a source of rewards; it also depends on the total number of iterations $n$ that the agent spends continuously at a source before leaving it, and on the switch-related travel cost $c_{sw}$ that an agent expends to travel between the reward sources. If all sources of reward in the task are identical (e.g. the "fixed initial rewards" task), the average reward intake rate $\mathfrak{R}$ matches the average over one visit to a reward source, and is expressed by the total reward (minus cost) divided by the number of steps:

$$\mathfrak{R} = \frac{1}{n} \sum_{m=1}^{n} \tilde{r}_m \equiv \frac{1}{n} (\sum_{m=1}^{n} r_m - c_{sw}) \tag{5}$$

For simplicity of notation, we define the cumulative sum of all received rewards gained throughout a stay at a source of rewards for $n$ sequential iterations as $\sum_{m=1}^{n} r_m = g(n)$:

$$\mathfrak{R} = \frac{1}{n} (g(n) - c_{sw}) \tag{6}$$

Similarly if the task contain multiple different types of reward sources (e.g. the "random initial rewards" task) offering cumulative rewards $g_i(n_i)$ and visited with probabilities $p_i$, the average reward intake rate would be given by the Equation 7. Here we imply that the agent spends a fixed number of steps $n_i$ at each reward source of type $i$, i.e. behaves consistently under the same circumstances.

$$\mathfrak{R} = \frac{\sum_i p_i g_i(n_i) - c_{sw}}{\sum_i p_i n_i} \tag{7}$$

To maximize the reward intake $\mathfrak{R}$, we calculate its partial derivative w.r.t the number of sequential steps $n_i$ at the current source of rewards. We then find the number of sequential iterations $n_i$ at this

source, which would turn the partial derivative of reward intake to zero. The partial derivative only affects those terms of the sums containing $n_i$. Below, we omit the denominator for brevity.

$$0 = \frac{\partial \mathfrak{R}}{\partial n_i} \sim (p_i \frac{\partial g_i}{\partial n_i})(\sum_i p_i n_i) - (p_i)(\sum_i p_i g_i - c_{sw}) \tag{8}$$

$$\frac{\partial g_i}{\partial n_i} = \frac{\sum_i p_i g_i}{\sum_i p_i n_i} - \frac{c_{sw}}{\sum_i p_i n_i} \tag{9}$$

The discrete derivative of cumulative reward $g_i$ received from one reward source, represented by the left-hand side of the Equation 9, can be estimated as the difference between the next and current cumulative rewards: $g_i(n + 1) - g_i(n) = r_{n+1}$. In the right-hand part of the Equation 9, the first term represents the average value of the reward $\langle r \rangle$ received by the agent in a single iteration in the environment. The second term here equals to the average switch-related travel cost $\langle c_{sw} \rangle$ computed over all trials with non-zero or zero switching cost. Overall, the Equation 9 can be rewritten as:

$$r_{n+1} = \langle r \rangle - \langle c_{sw} \rangle \tag{10}$$

$$\tilde{r}_{n+1} = \langle \tilde{r} \rangle \tag{11}$$

which proves the theorem. This result does not depend on the particular sequence of the rewards $r_n$, but only requires the rewards do be decreasing over time. □

## A.2 The V-learning optimum for the "fixed initial rewards" task

**Theorem 2** (the V-learning optimum). *The optimal policy for a V-learning agent navigating between sources of depleting reward in the limit of low temporal discount $\gamma \to 1$ is defined by the MVT, i.e. the agent should leave a source of depleting rewards when the next expected reward $\tilde{r}_{n+1}$ falls below the average reward in the environment $\langle \tilde{r}_n \rangle$ computed over rewards $\{r\}$ and travel costs $\{c_{sw}\}$.*

*Proof.* The objective of the V-learning is to maximize the cumulative future discounted reward:

$$V_t = \mathbb{E} \sum_\tau \tilde{r}_{t+\tau} \gamma^\tau \tag{12}$$

The reward $\tilde{r}_{t+\tau}$ here can be decomposed into its average value $\langle \tilde{r} \rangle$ and the residual part $\Delta \tilde{r}_{t+\tau}$:

$$V_t = \mathbb{E} \sum_\tau (\langle \tilde{r} \rangle + \Delta \tilde{r}_{t+\tau}) \gamma^\tau \tag{13}$$

As the average reward value $\langle \tilde{r} \rangle$ within the sum is constant, we can sum its corresponding coefficients $\gamma^\tau$ as a geometric progression:

$$V_t = \frac{1}{1 - \gamma} \mathbb{E} \langle \tilde{r} \rangle + \mathbb{E} \sum_\tau (\Delta \tilde{r}_{t+\tau}) \gamma^\tau \tag{14}$$

In the limit of low temporal discount $\gamma \to 1$, the first term of the right-hand part of (14) is large (infinitely large in the limit of $\gamma = 1$), whereas the second term aims zero:

$$\mathbb{E} \sum_\tau \Delta \tilde{r}_{t+\tau} = \sum_\tau \mathbb{E} \Delta \tilde{r}_{t+\tau} = 0 \tag{15}$$

The second term (15) of the Equation 14 in the limit of $\gamma \to 1$ is negligible compared to the first term. Therefore, maximization of the V-function is equivalent to maximization of the first term in (14), reflecting the expected average reward intake $\langle \tilde{r} \rangle$ per single iteration, which is equivalent to the expected average reward intake *rate*. The optimal policy for maximizing the reward intake rate is described by the Theorem 1 (the MVT), which proves the theorem. □

**Corollary 2.1** (the V-learning optimum for the "fixed initial rewards" task). *The optimal V-learning agent navigating between reward sources offering the initial reward $r_0$ which depletes exponentially at the rate $\xi$ s.t. $r_{n+1} = \xi r_n$, in the limit of low temporal discount $\gamma \to 1$ pays the switching-related travel cost $c_{sw}$ and leaves a reward source after $n$ trials where $n$ is defined by the following equation:*

$$r_0 = c_{sw} \left( \frac{1 - \xi^n}{1 - \xi} - n\xi^n \right)^{-1} \tag{16}$$

*Proof.* Let $r_n$ be the current reward, corresponding to sequential iteration $n$ within a reward source; $\langle r_n \rangle$ – a flat average reward after $n$ sequential iterations within current source of rewards. In accordance with the Theorem 2, the optimal V-learning agent leaves the reward source when the next expected reward $r_{n+1}$ at the source falls below the average reward in the environment $\langle r_n \rangle$ discounted by the travel costs $\langle c_{sw} \rangle$:

$$\langle r_n \rangle - \langle c_{sw} \rangle = r_{n+1} \tag{17}$$

Due to the exponential depletion of rewards within a source at a rate $\xi$, current reward $r_n$ on sequential iteration $n$ at the same source can be expressed as $r_n = \xi^{n-1} r_0$. The Equation 17 can be rewritten:

$$\frac{1}{n} \left( \sum_{k=1}^{n} (\xi^{k-1} r_0) - c_{sw} \right) = \xi^n r_0 \tag{18}$$

The sum in the left-hand side of the Equation 18, used in calculation of the average reward $\langle r_n \rangle$, can be evaluated as a sum of the geometric progression:

$$\frac{1}{n} \left( r_0 \frac{1 - \xi^n}{1 - \xi} - c_{sw} \right) = \xi^n r_0 \tag{19}$$

We aggregate all the terms containing the initial reward value $r_0$ at the reward source to obtain the relation between $r_0$ and $n$, the number of sequential trials that an optimal V-learning agent spends at a reward source before leaving it:

$$r_0 = c_{sw} \left( \frac{1 - \xi^n}{1 - \xi} - n\xi^n \right)^{-1} \tag{20}$$

which proves the corollary. The Equation 20 cannot be inverted analytically without use of special functions. The optimal number of sequential iterations $n(r_0)$ that a V-learning agent is expected to spend at a reward source before leaving it can be found as a numerical inverse of (20). □

### A.3 The R-learning optimum for the "fixed initial rewards" task

**Theorem 3** (the R-learning optimum). *The optimal policy for an R-learning agent navigating between sources of depleting reward in the limit of low temporal discount $\gamma \to 1$ is to leave a source of depleting reward when the next expected reward $\tilde{r}_{n+1}$ at the source falls below an exponential average $\tilde{r}_n^{avg}$ of the previous rewards and travel costs – the decision rule we called the **Leaky MVT**:*

$$\tilde{r}_{n+1} = \tilde{r}_n^{avg} \equiv (1 - \kappa) \sum_{m=0}^{\infty} \kappa^m \tilde{r}_{n-m} \tag{21}$$

*Proof.* Assume an alternative, *virtual* reward schedule $\tilde{r}^*$ corresponding to the perceived reward $\tilde{r}$ as follows: the *virtual* reward of this agent $\tilde{r}_n^*$ (i.e. the quantity optimized by the agent) is equal to the difference between the reward $\tilde{r}_n$, and its exponential average over past trials $\tilde{r}_{n-1}^{avg}$:

$$\tilde{r}_n^* = \tilde{r}_n - \tilde{r}_{n-1}^{avg} \tag{22}$$

The reward $\tilde{r}_n$ is an exponentially depleting value. The exponentially averaged reward $\tilde{r}_{n-1}^{avg}$ is a slowly changing variable compared to the reward $\tilde{r}_n$ if its exponential averaging rate $\kappa$ exceeds the reward depletion rate $\xi$: ($\kappa > \xi$). As a result, the agent's *virtual* reward, being a difference

of the fast-changing reward $\tilde{r}_n$ and a slow-changing exponentially averaged reward $\tilde{r}_{n-1}^{avg}$ is also approximately a depleting value. Consequently, we can apply the Theorem 2 to maximize the average intake rate of the effective reward $\tilde{r}^*$ that an agent collects in the environment:

$$\tilde{r}_{n+1} - \tilde{r}_n^{avg} = \langle \tilde{r}_n - \tilde{r}_{n-1}^{avg} \rangle \tag{23}$$

We then calculate the right-hand side of the decision rule (23). The flat average of the exponential average of the rewards can be represented as the average of convolution between the vector of exponential averaging coefficients $\vec{\kappa} = (1-\kappa) \cdot (\kappa, \kappa^2, \kappa^3, ...)$ and the vector of rewards $\vec{\tilde{r}} = (\tilde{r}_m, \tilde{r}_{m-1}, \tilde{r}_{m-2}, ...)$. Here $(1-\kappa)$ normalizes the exponential averaging coefficients so that, on the average, the exponential average matches the flat average of rewards.

$$\langle \tilde{r}_n - \tilde{r}_{n-1}^{avg} \rangle = \langle \tilde{r} \rangle - \langle \text{conv}(\vec{\kappa}, \vec{\tilde{r}}) \rangle \tag{24}$$

Due to the properties of convolution, its average $\langle \text{conv}(a, b) \rangle$ equals to the product of $||a||_1$ and $\langle b \rangle$. As the exponential average is normalized so that $||\vec{\kappa}||_1 = 1$, the average reward value $||\vec{\kappa}||_1 \langle \vec{\tilde{r}} \rangle = \langle \tilde{r} \rangle$ cancels out, the right-hand side of the decision rule (23) can be written as:

$$\tilde{r}_{n+1} = \tilde{r}_n^{avg} \tag{25}$$

As $\tilde{r}^*$ replaces the reward in R-learning, the rule described by the Equation 23 optimizes it. $\qquad\square$

**Corollary 3.1** (the R-learning optimum for the "fixed initial rewards" task)**.** *The optimal R-learning agent, which averages rewards exponentially at the rate $\kappa$, navigating between reward sources offering the initial reward $r_0$ which depletes exponentially at the rate $\xi$ s.t. $r_{n+1} = \xi r_n$, in the limit of low temporal discount $\gamma \to 1$ pays the switching-related travel cost $c_{sw}$ and leaves a reward source after $n$ sequential iterations where $n$ is defined by the following equation. In the limit of $\kappa \to 1$, it matches the equation for V-learning.*

$$r_0 = \kappa^{n-1} c_{sw} \left( \frac{\kappa^n - \xi^n}{\kappa - \xi} - \frac{1 - \kappa^n}{1 - \kappa} \xi^n \right)^{-1} \tag{26}$$

*Proof.* Using the definition (21) of the exponential averaging at a rate $\kappa$ stating that $\tilde{r}_{n+1}^{avg} = \kappa \tilde{r}_n^{avg} + (1-\kappa)\tilde{r}_{n+1}$, we write the expression for the exponentially averaged reward value $\tilde{r}_1^{avg}$ after an agent received the first reward at a new source. We assume that the agent spends $n$ sequential trials at each source of reward before leaving, and call the average reward at the moment of leaving $\tilde{r}_n^{avg}$ (to be determined later):

$$\tilde{r}_1^{avg} = \kappa \tilde{r}_n^{avg} + (1-\kappa)(r_0 - c_{sw}) \tag{27}$$

Similarly, we can write the expression for the exponential average reward $\tilde{r}_2^{avg}$ after 2 iterations at a new source of rewards, and then generalize it to the case $\tilde{r}_m^{avg}$ of $m$ sequential iterations at a source:

$$\tilde{r}_2^{avg} = \kappa \left( \kappa \tilde{r}_n^{avg} + (1-\kappa)(r_0 - c_{sw}) \right) + (1-\kappa)\xi r_0 \tag{28}$$

$$\tilde{r}_2^{avg} = \kappa^2 \tilde{r}_n^{avg} + \kappa(1-\kappa)(r_0 - c_{sw}) + (1-\kappa)\xi r_0 \tag{29}$$

$$\tilde{r}_m^{avg} = \kappa^m \tilde{r}_n^{avg} + (1-\kappa) \left( \sum_{k=0}^{m-1} \kappa^{m-1-k}\xi^k r_0 - \kappa^{m-1} c_{sw} \right) \tag{30}$$

We further extract the common multiplier $\kappa^{m-1}$ out of sum in the right-hand part of (30), and calculate the sum of the remaining geometric progression:

$$\tilde{r}_m^{avg} = \kappa^m \tilde{r}_n^{avg} + (1-\kappa) \left( \kappa^{m-1} \sum_{k=0}^{m-1} (\xi/\kappa)^k r_0 - \kappa^{m-1} c_{sw} \right) \tag{31}$$

$$\tilde{r}_m^{avg} = \kappa^m \tilde{r}_n^{avg} + (1-\kappa) \left( \frac{\kappa^m}{\kappa} \frac{1 - (\xi/\kappa)^m}{1 - (\xi/\kappa)} r_0 - \kappa^{m-1} c_{sw} \right) \tag{32}$$

$$\tilde{r}_m^{avg} = \kappa^m \tilde{r}_n^{avg} + (1 - \kappa)\left(\frac{\kappa^m - \xi^m}{\kappa - \xi} r_0 - \kappa^{m-1} c_{sw}\right) \tag{33}$$

We now focus on the iteration when the agent has to leave the source of rewards again, i.e. $m = n$. This iteration, considered in the Equation 33, allows us to define the average reward $\tilde{r}_n^{avg}$ at the moment of leaving a reward source

$$\tilde{r}_n^{avg} = \frac{1 - \kappa}{1 - \kappa^n}\left(\frac{\kappa^n - \xi^n}{\kappa - \xi} r_0 - \kappa^{n-1} c_{sw}\right) \tag{34}$$

We assert that in the limit of long-time exponential averaging ($\kappa \to 1$) the result in the Equation 34 matches the flat average of the rewards as used in (19), being the sum of geometric progression of rewards $\sum_{m=1}^{n} \xi^{m-1} r_m$ divided by $n$. We then plug the identified expression (34) for exponential average of rewards at the moment of leaving a source $\tilde{r}_n^{avg}$ into the right-hand side of the Leaky MVT (21), providing the optimal decision rule for R-learning predicted by the Theorem 3:

$$r_0 \xi^n \frac{1 - \kappa^n}{1 - \kappa} = \frac{\kappa^n - \xi^n}{\kappa - \xi} r_0 - \kappa^{n-1} c_{sw} \tag{35}$$

$$r_0 = \kappa^{n-1} c_{sw}\left(\frac{\kappa^n - \xi^n}{\kappa - \xi} - \frac{1 - \kappa^n}{1 - \kappa}\xi^n\right)^{-1} \tag{36}$$

which proves the corollary. Like in the case of Corollary 2.1, the Equation 36 cannot be inverted analytically without use of special functions. The optimal number of sequential iterations $n(r_0)$, which an R-learning agent is expected to spend at a reward source before leaving, can be found as a numerical inverse of (36). $\qquad\square$

### A.4 The R- (and V-) learning optima for the "random initial rewards" task

**Corollary 3.2** (the R-learning optimum for the "random initial rewards" task). *The optimal R-learning agent, which averages rewards exponentially at the rate $\kappa$, navigating between reward sources offering random initial reward $r_0 \sim \{r_0^1, ..., r_0^k\}$ which deplete exponentially at the rate $\xi$ s.t. $r_{n+1} = \xi r_n$, in the limit of low temporal discount $\gamma \to 1$ pays the switching-related travel cost $c_{sw}$ and leaves a reward source at the reward value $r_{sw}$ which grows with $r_0$ if exponential averaging is finite ($\kappa < 1$) and if forgetting is slower than the reward decay ($\xi < \kappa$):*

$$sgn(r_{sw} - \langle r_{sw}\rangle) = sgn\left((\log r_0 - \log\langle r_0\rangle)(1 - \kappa)\right) \tag{37}$$

*Proof.* To proceed with this derivation, first consider a case where a single novel restart value has been introduced after a prolonged period of the "fixed initial rewards" schedule.

The left-hand side of the Leaky MVT decision rule (25) would not change. On the contrast, the right-hand side of the Leaky MVT has to be rewritten using the general case expression for the exponentially averaged reward (33), keeping in mind that until now the exponentially averaged reward satisfied the stable-state expression (34). Here: $r_0^{new}$ – a novel surprising restart value; $m$ – the number of iterations from $r_0^{new}$ to the current reward value.

$$\tilde{r}_m^{avg} = \kappa^m\left(\frac{1 - \kappa}{1 - \kappa^n}\left(\frac{\kappa^n - \xi^n}{\kappa - \xi} r_0 - \kappa^{n-1} c_{sw}\right)\right) + (1 - \kappa)\left(\frac{\kappa^m - \xi^m}{\kappa - \xi} r_0^{new} - \kappa^{m-1} c_{sw}\right) \tag{38}$$

$$r_0^{new}\xi^m - \kappa^m\left(\frac{1 - \kappa}{1 - \kappa^n}\left(\frac{\kappa^n - \xi^n}{\kappa - \xi} r_0 - \kappa^{n-1} c_{sw}\right)\right) - (1 - \kappa)\left(\frac{\kappa^m - \xi^m}{\kappa - \xi} r_0^{new} - \kappa^{m-1} c_{sw}\right) = 0 \tag{39}$$

A similar expression to (39) can be written for the usual case when the initial reward at a new source of rewards is not surprising, leading to the usual leaving of the new reward source after $m = n$ sequential iterations:

$$r_0\xi^n - \kappa^n\left(\frac{1 - \kappa}{1 - \kappa^n}\left(\frac{\kappa^n - \xi^n}{\kappa - \xi} r_0 - \kappa^{n-1} c_{sw}\right)\right) - (1 - \kappa)\left(\frac{\kappa^n - \xi^n}{\kappa - \xi} r_0 - \kappa^{n-1} c_{sw}\right) = 0 \tag{40}$$

Now consider the Equations (39) and (40) at the same level of current reward ($r_0\xi^n = r_0^{new}\xi^m \equiv r$). We are interested in the difference between the Equations 39 and 40. This way, we will see whether the agent will leave the reward sources at the same level of current reward. For instance, if the difference between the Equations (39) and (40) is positive, it would mean that the left-hand side of (39) is larger than the right-hand side; the Leaky MVT decision rule criterion therefore has not been met at the reward $r$ yet, and the agent will stay longer at the same reward source.

$$(\kappa^n - \kappa^m)\left(\frac{1-\kappa}{1-\kappa^n}\left(\frac{\kappa^n - \xi^n}{\kappa - \xi}r_0 - \kappa^{n-1}c_{sw}\right)\right) -$$
$$-(1-\kappa)\left(\frac{r_0^{new}\kappa^m - r_0\kappa^n}{\kappa - \xi} - (\kappa^{m-1} - \kappa^{n-1})c_{sw}\right) > 0 \tag{41}$$

$$(\kappa^n - \kappa^m)\left(\frac{\kappa^n - \xi^n}{1-\kappa^n}r_0 - \frac{\kappa - \xi}{1-\kappa^n}\kappa^{n-1}c_{sw}\right) - (r_0\xi^{n-m}\kappa^m - r_0\kappa^n) + (\kappa - \xi)(\kappa^{m-1} - \kappa^{n-1})c_{sw} > 0 \tag{42}$$

Here we have omitted two multipliers, $(1-\kappa)$ and $(\kappa - \xi)^{-1}$, which are both positive under our assumptions ($\xi < \kappa < 1$). In general, however, these multipliers are not necessarily positive, so we will reintroduce them later.

$$\frac{\kappa^n - \kappa^m}{1-\kappa^n}(\kappa^n - \xi^n) - \left(\left(\frac{\xi}{\kappa}\right)^{n-m} - 1\right)\kappa^n - (\kappa^n - \kappa^m)(\kappa - \xi)\frac{c_{sw}}{r_0}\left(\frac{\kappa^{n-1}}{1-\kappa^n} + \frac{1}{\kappa}\right) > 0 \tag{43}$$

$$-\frac{\kappa^n - \kappa^m}{1-\kappa^n}\left(\left(\frac{\xi}{\kappa}\right)^n - 1\right) - \left(\left(\frac{\xi}{\kappa}\right)^{n-m} - 1\right) - \frac{(\kappa^n - \kappa^m)(\kappa - \xi)}{(1-\kappa^n)\kappa}\frac{c_{sw}}{r_0} > 0 \tag{44}$$

In the experiments, both $\kappa$ and $\xi/\kappa$ are approximately equal to 0.9, which allows us to use the 2nd order Taylor expansion w.r.t these variables:

$$-\frac{m-n}{n}\left(-n\left(1-\frac{\xi}{\kappa}\right) + \frac{1}{2}n(n-1)\left(1-\frac{\xi}{\kappa}\right)^2\right) -$$
$$-\left(-(n-m)\left(1-\frac{\xi}{\kappa}\right) + \frac{1}{2}(n-m)(n-m-1)\left(1-\frac{\xi}{\kappa}\right)^2\right) - \frac{m-n}{n}\left(1-\frac{\xi}{\kappa}\right)\frac{c_{sw}}{r_0} > 0 \tag{45}$$

We simplify the Equation (45) while using the notion of the average switching cost $\langle c_{sw}\rangle \equiv c_{sw}/n$:

$$-\left(\frac{\langle c_{sw}\rangle}{r_0} + \frac{m}{2}\left(1-\frac{\xi}{\kappa}\right)\right)(m-n)\left(1-\frac{\xi}{\kappa}\right) > 0 \tag{46}$$

As $m$ and $n$ are the numbers of steps from $r_0^{new}$ and $r_0$ respectively to $r$, these values can be rewritten in terms of the initial and current rewards, keeping in mind the exponential reward schedule:

$$m - n = \log_\xi \frac{r}{r_0^{new}} - \log_\xi \frac{r}{r_0} = \log_\xi \frac{r_0}{r_0^{new}} \sim -\log \frac{r_0}{r_0^{new}} \tag{47}$$

$$\left(\frac{\langle c_{sw}\rangle}{r_0} + \frac{m}{2}\left(1-\frac{\xi}{\kappa}\right)\right)\log\frac{r_0}{r_0^{new}} \cdot \left(1-\frac{\xi}{\kappa}\right) > 0 \tag{48}$$

As a result, if $r_0^{new}$ is smaller than $r_0$, the quantity in (48) is positive and the switch will occur later in the reward sequence, at the reward level smaller than $r$. To consider the effect of the other parameters on stay-or-leave decisions, we bring back the multipliers $(1-\kappa)$ and $(\kappa - \xi)^{-1}$, omitted in (42):

$$\left(\frac{\langle c_{sw}\rangle}{r_0} + \frac{m}{2}\left(1-\frac{\xi}{\kappa}\right)\right)\log\frac{r_0}{r_0^{new}} \cdot \left(1-\frac{\xi}{\kappa}\right)\frac{1-\kappa}{\kappa - \xi} > 0 \tag{49}$$

Below we assess the relative contribution of both terms in the first parentheses of the equation (49) above. The switching cost, as estimated from the data fits, roughly equals to $c_{sw} = 4$; the average

restart reward was equal to $r_0 = 12$, and the average number of trials before switching a port was equal to $m = 4$. The reward decrease coefficient was selected to be $\xi = 0.8$, and the exponential averaging rate was estimated from the data as $\kappa = 0.9$ This way, the terms in the first parentheses of (49) are roughly equal to 0.1 and 0.2 respectively, implying that exponential forgetting of rewards and switching costs in the R-learning / Leaky MVT model contribute comparably to stay-or-leave decisions. We further discard both of these coefficients as their sum in parentheses is positive and does not affect the sign in (49):

$$\log \frac{r_0}{r_0^{new}} \cdot (1 - \kappa) > 0 \tag{50}$$

The logic from this paragraph can be extended to the case where each restart value is selected randomly. In this paragraph, the key idea of a single surprising restart value was that we know the average reward value $\tilde{r}_n^{avg}$. Now, if we use randomly selected restart rewards on every restart, we can on average consider these restart values as one-time surprising values w.r.t some average-case policy, starting from the reward $\langle r_0 \rangle^*$, and lasting for $\langle n \rangle^*$ iterations. These randomly selected restart values therefore would not on average change the average-case constants, such as $\langle \tilde{r}_n^{avg} \rangle^*$, allowing us to use the logic from this paragraph and to extend its predictions to the "random initial rewards" task:

$$\log \frac{\langle r_0 \rangle}{r_0} \cdot (1 - \kappa) > 0 \tag{51}$$

$$\mathrm{sgn}(r_{sw} - \langle r_{sw} \rangle) = \mathrm{sgn}\left((\log r_0 - \log\langle r_0 \rangle)(1 - \kappa)\right) \tag{52}$$

which proves the corollary. $\qquad\square$

**Corollary 3.3** (the V-learning optimum for the "random initial rewards" task)**.** *The optimal V-learning agent, navigating between reward sources offering random initial rewards $r_0 \sim \{r_0^1, ..., r_0^k\}$, which deplete exponentially at the rate $\xi$ s.t. $r_{n+1} = \xi r_n$, in the limit of low temporal discount $\gamma \to 1$ pays the switching-related travel cost $c_{sw}$ and leaves a reward source at the same reward value $r_{sw}$ regardless of $r_0$.*

*Proof.* V-learning may be viewed as R-learning with infinitle exponential averaging ($\kappa \to 1$). In the Equation 52, $\kappa = 1$ turns the right-hand side to zero implying that the reward threshold at which an agent would leave a reward source $r_{sw}$ would be equal to its average value $\langle r_{sw} \rangle$ independently of random initial reward $r_0$. $\qquad\square$

## A.5   Inference of the model parameters for MVT and Leaky MVT; model comparison

To infer the model parameters for MVT and Leaky MVT decision rules from our mouse data, we follow the approach used in logistic regression. The probability $p$ to correctly predict the animal's actions can be written as a function of the animal's recorded choices $t$ and predicted choices $y$:

$$p(t|w) = \prod_{n=1}^{N} y_n^{t_n}(1 - y_n)^{1-t_n} \text{ s.t. } y_n \equiv \sigma(w[\phi_n]) \tag{53}$$

Here $w$ are the functions computed on the model's inputs $\phi$ and optimized so that the model yields the best predictions. The best predictions are defined so that they optimize the negative log likelihood $L$ of the model to predict the animal's recorded actions:

$$L = -\log p(t|w) = -\sum_{n=1}^{N} (t_n \log y_n + (1 - t_n) \log(1 - y_n)) \tag{54}$$

The negative log likelihood $L$ can be optimized by setting its gradient to zero:

$$\nabla_w L = -\sum_{n=1}^{N} \left( t_n \frac{1}{y_n} y_n(1 - y_n) + (1 - t_n)\frac{1}{1 - y_n} y_n(1 - y_n) \right) \frac{\partial w[\phi]}{\partial w} \tag{55}$$

$$\nabla_w L = -\sum_{n=1}^{N} (y_n - t_n)\frac{\partial w[\phi]}{\partial w} \tag{56}$$

In case of the MVT (Equation 11), the decision variable for a stay-or-leave action is computed as $w[\phi] \equiv w[r_{t+1}] = \alpha \cdot (r_{t+1} - \langle \tilde{r} \rangle) \equiv \alpha \xi r_t - \alpha \langle \tilde{r} \rangle \equiv w_1 r_t + w_2$ (here $\alpha$ is the decision noise). The parameters $w_1$ and $w_2$ affect the model inputs $r_t$ and 1 independently:

$$\nabla_{w_1} L = -\sum_{n=1}^{N}(y_n - t_n)r_t; \nabla_{w_2} L = -\sum_{n=1}^{N}(y_n - t_n) \tag{57}$$

The equation above corresponds to the classic format of logistic regression: it has an exact solution, or can be solved approximately using gradient descent. In case of the the Leaky MVT (Equation 21), the decision variable is computed as $w[\phi] = \alpha \cdot (\tilde{r}_{t+1} - \tilde{r}_t^{avg}) \equiv \alpha \cdot (\xi r_t - (1-\kappa)\sum_{\tau=0}^{\infty} \kappa^\tau (r_{t-\tau} - c_{sw}\delta_{t-\tau}^{sw}))$ where delta-function $\delta_{t-\tau}^{sw}$ indicates the trials to which the switching cost has been applied. To optimize the log likelihood $L$, we compute its derivatives w.r.t the model parameters $(\alpha, \kappa, c_{sw})$:

$$\frac{\partial w[\vec{r}, \vec{\delta}^{sw}]}{\partial \alpha} = \xi r_t - \tilde{r}^{avg} \tag{58}$$

$$\frac{\partial w[\vec{r}, \vec{\delta}^{sw}]}{\partial c_{sw}} = \alpha(1-\kappa)\sum_{\tau=0}^{\infty} \kappa^\tau \delta_{t-\tau}^{sw} \tag{59}$$

$$\frac{\partial w[\vec{r}, \vec{\delta}^{sw}]}{\partial \kappa} = \alpha \frac{\tilde{r}^{avg}}{1-\kappa} - \alpha(1-\kappa)\sum_{\tau=0}^{\infty} \tau \kappa^{\tau-1} \tilde{r}_{t-\tau} \tag{60}$$

To compute the gradient of the log likelihood, we put partial derivatives above into the Equation 56:

$$\nabla_\alpha L = -\sum_{n=1}^{N}(y_n - t_n)(\xi r_n - \tilde{r}_n^{avg}) \tag{61}$$

$$\nabla_{c_{sw}} L = -\alpha(1-\kappa)\sum_{n=1}^{N}(y_n - t_n)\sum_{\tau=0}^{\infty} \kappa^\tau \delta_{n-\tau}^{sw} \tag{62}$$

$$\nabla_\kappa L = -\alpha \sum_{n=1}^{N}(y_n - t_n)\left(\frac{\tilde{r}_n^{avg}}{1-\kappa} - (1-\kappa)\sum_{\tau=0}^{\infty} \tau \kappa^{\tau-1} \tilde{r}_{n-\tau}\right) \tag{63}$$

As the model parameters $(\alpha, \kappa, c_{sw})$ are not independent, we cannot derive the exact solution as for the linear system in case of the MVT, so we have to run the gradient descent instead. To perform an adequate comparison between the MVT and Leaky MVT models, we therefore find optima for each of them using gradient descent under the same learning parameters. We then use the Akaike information criterion (AIC) to establish which model explains the data better:

$$\text{AIC} = 2(k + L) \tag{64}$$

where the number of parameters $k$ is equal to $k = 2$ for the MVT, and $k = 3$ for the Leaky MVT, and $L$ is the negative log likelihood of a model to predict the recorded animal's choices.