[Reviews · NeurIPS 2020]

Review 1

Summary and Contributions: The authors collected behavioral and neural data from mice performing two foraging tasks. They compared R-learning and V-learning model on the data, and found that R-learning can explain the behavioral data better, while the neural data is consistent with the TD error of both models. They also showed analytically how their model is related to the optimal policy, and extensively discussed how their work can be related to some other work in the field.

Strengths: The paper is a good combination of neuroscience, machine learning, and theory, which I found all of them are solid and nicely done. I think it’s relevant to the board audience of NeurIPS.

Weaknesses: It would be good to compare some other simple, non-deep baseline models, and show if the experimental finding is only recoverable by deep R-learning. The neural data is consistent with both models, so it does not strongly support R-learning. It would be interesting to see if there’s any neural substrate that strongly supports R-learning but not the other models. The authors claim that their work links RL, MVT, and Bayesian inference approach. It was clear how their work link RL and MVT, but I wasn’t sure how it is related to the Bayesian inference approach. The author did discuss a Bayesian model in the discussion, but it would be good to maybe perform model comparison. --- updates --- I appreciate authors' effort into addressing most of my concerns. so I'm increasing my score. The clarity of the paper is good, but if accepted, it would be good to have one more iteration on the clarity. And the results of alternative baseline models can all go into the supplementary material.

Correctness: They are correct to the best of my knowledge.

Clarity: The paper is very well-written and I found it easy to follow. I appreciate the effort. I think R-learning first appears in line 73 without any introduction. Is the R-learning using in this paper the same as Schwartz, 1993? Then I think it would be good to cite it at the first appearance, and with a brief introduction. It wasn’t clear how you model the switching cost in your model. The training schedule is not very clear to me. For example, did you train the model separately for different initial values in task 1? And did you train the model separately for task 1 and task 2? Line 34-36 in the supplementary material seems to indicate some sequential training, that you train the model on task 1, and then on task 2, is that right? Line 301, “is” -> “in” dynamic real-world environment

Relation to Prior Work: The authors mentioned in the paper that their task 2 is a novel experimental design, which is not. For example, in Constantino & Daw, 2015, which is cited by the authors, they also have the same initial values in experiment 1, and a random initial value in experiment 2. So it wasn’t clear to me the novelty of the experimental design. And it wasn’t clear if their models are novel or established, and how their results and methods compared to Constantino & Daw, 2015. It would be interesting if the authors can further discuss experimental and modeling results compared to Constantino & Daw, 2015, which they found that human subjects overharvesting and also considered an R-learning model and other TD models.

Reproducibility: Yes

Additional Feedback: It is an interesting finding that the mice under-harvesting the current resource, do you think it is related to the short travel time between port, i.e. the switching cost is relatively low? The authors discussed that the R-learning model could offer a mechanism for learning to make these decisions (line 269-175). I agree that based on the findings of this paper, it seems R-learning captures animal paradoxical escape better, but is this behavior beneficial? It is valuable to understand animal behavior, but it might not be valuable to implement such model in the real-world application if the model leads to less accumulated reward. There are some claims in the discussion that seems to be not well-supported, e.g. line 298-300, and line 317-320. It is nice to provide insights and speculations, but it would be good if the authors can make it clear whether it is already supported by any result, or it is a possible hypothesis for future work. Also, the author noted that the TD errors in V- and Q-learning are different during training before they converge to an optimal policy. It would be interesting to look at neural recordings during training.


Review 2

Summary and Contributions: This is an impressive and timely study that suggests a reference point-based framework for (deep) reinforcement learning and shows how various results from rodent foraging tasks could be reproduced better than using classical RL approaches. The paper provides a highly desirable link between behavioural economics and reinforcement learning communities and is strong on both theoretical and empirical aspects. However, more could be done to tease out the differences between V and R learning (especially with regard to VTA activity) and integrate with results with prospect theory/reference point literature, which is vast, even if it does not normally employ RL. More attention should also be paid for parameter estimation and their interpretation.

Strengths: The paper provides both theoretical and empirical contributions, connects RL with behavioural economics (more implicitly than explicitly) and provides some neural evidence that VTA responses are not inconsistent with R-learning model predictions. The findings have huge potential of significance to computational cognitive neuroscience community and to NeurIPS. Although the idea of reference point-based valuation is not novel, the theoretical framework of R learning is (even if reward rate has been used in models of effort or motivation).

Weaknesses: More attention should be paid for teasing out differences between V and R learning, with intermittent initial rewards being essentially the only example. Although it is impressive that new VTA recording data is presented in the paper, I don't feel that the result is particularly helpful - it only shows that VTA activity doesn't contradict R-learning model, but it does not really provide specific support for it. It should be possible to design different tasks/protocols under which the two formalisations would have substantially different TD errors, which could help tease out biological correlates of the two models. Furthermore, it would be nice to see more details of parameter estimation and the resulting best-fitting parameter values, which if done properly, may allow to achieve not only a qualitative but also a better quantitative fit between Fig. 1E and Fig. 3D (as well as between Fig. 1D and Fig. 3B). As the models have multiple parameters substantially affecting performance, the two models should be compared under best-fitting parameters and should include formal measures like AIC, not just qualitative fits. Of course model universality regardless of parameters is helpful, but quantitative fit is equally important. Finally integration with behavioural economics literature should be improved.

Correctness: Although I didn't check details of all proofs in supplementary material, the results look intuitive and correct, and the overall methodology is sound.

Clarity: The paper is clearly written and only has a few typos (in lines 302 and 305).

Relation to Prior Work: Although relation to prior work is discussed reasonably well, there could be better integration with behavioural economics and neuroeconomics literature on reference point-based valuation, which is pretty vast. It's also important to point out that the overall idea of reference-based valuation is not novel, although the theoretical framework is. It would also help mentioning effort and motivation models that use similar formalisms and are relevant here (e.g. animals may switch faster due to boredom or fail to switch due to increasing fatigue). I also think that discussion is somewhat repetitive and the saved space could be used to expand on my suggested aspects. Although discussion of connection to model-based vs model-free RL is helpful, the authors could go even further to suggest that the model-based vs. model-free dichotomy is unnecessary and should be avoided, especially with the help of the results and theoretical framework developed in this study, but not only (e.g. see https://papers.nips.cc/paper/3311-hippocampal-contributions-to-control-the-third-way.pdf )

Reproducibility: Yes

Additional Feedback: I thank the authors for promising to address my concerns such as quantitative fits and integration with behavioural economics literature. I hope motivation models could be mentioned as well, as they are also relevant. I agree that the recordings result is still useful, it's just important to point out its limitations.


Review 3

Summary and Contributions: This paper shows that R-learning models animal foraging behavior better than V-learning in a novel task where rewards per patch deplete over time and the animal must decide whether to stay or switch to a different patch. A further experiments showed that the behavioral prescription of the marginal value theorem fits better when you work it out for R learning than for V learning. In their final experiment, they recorded from dopaminergic neurons in VTA, but apparently found either V or R learning explained the results equally well as one other.

Strengths: The experiments are well designed and clearly explained.

Weaknesses: I remain unconvinced that the difference between whether neurons are better fit by the TD errors of V learning or the TD errors of R learning is terribly significant for neuroscience. Surely the neurons do not actually compute either quantity. These are just convenient abstractions. Maybe there is something I'm missing about the theoretical importance of R learning that the authors could help me to see with their reply?

Correctness: yes

Clarity: yes

Relation to Prior Work: yes

Reproducibility: Yes

Additional Feedback: Thanks for the reply to my question in the rebuttal. I found it helpful.


Review 4

Summary and Contributions: This paper uses a combination of experimental and analytical analysis to study the behaviour of animals in foraging tasks. The main point made by the paper is that when the initial reward of available resources is random, animals tend to show a sub-optimal behaviour and leave higher reward resources sooner than the others. The authors then use a combination of model simulation and analysis to explain this behaviour.

Strengths: I enjoyed reading the paper. It is well-written and provides interesting insights into the potential explanation of animal’s behaviour in foraging tasks. I also liked the analytical approach taken here to explore general properties of the model.

Weaknesses: The effects in Figure 3D are very small and different from the data reported in Figure 1E. In particular, the threshold to leave the port seems very similar across the conditions in Figure 1E. Although the neural data are not inconsistent with R-learning, they don’t support R-learning either. In fact, based on the high correlation between the error signals in R and V learning, it seems that the average reward is almost constant, which is inconsistent with the proposal here about the role of average reward in explaining the behaviour in the intermittent initial rewards. ============ after rebuttal ==================== The authors didn't comment on the match between model and data, so I still have the concern based on which I'll keep the score.

Correctness: I haven't checked all the proofs in supplementary materials, but had a high level look and they look sensible to me.

Clarity: Yes

Relation to Prior Work: Yes

Reproducibility: Yes

Additional Feedback: Minor: In Equation 9 a “sign” operator is missing on the right side. It is mentioned that the error signal is used to train the network, but it is unclear how the error signal was used to train the network (specially the actor network).

[Author Response · NeurIPS 2020]

We thank the reviewers for their time and thorough comments, as well as their valuation of our work including its relevance for NeurIPS. We will fix the typos, expand the references, further separate results from hypotheses, and fill in the details suggested by the reviewers (e.g. switching cost is subtracted from the reward after switch; training for each set of restart values is performed separately). For the larger discussion items, please find the detailed comments below.

**Alternative models, experiments and analyses.** We agree that several claims about the R-model would be strengthened by including formal comparison across models. To this end, we performed additional manipulations of animal task parameters (inter-trial delays; reward depletion rates; switch costs) and tested non-deep models (port/action-value V/Q-learning). These manipulations support the R-learning model and we briefly mention them in the paper (lines 270-275), but we did not include the corresponding data/details to meet the space constrains and keep the text clarity. Additionally, the reviewers highlighted the importance of quantitative fits. We agree and have performed these analyses that will be included in the revised paper. Briefly, we used gradient descent to minimize the negative log likelihood w.r.t the parameters of the MVT / leaky MVT – the decision rules, as we prove, optimal for the V/R models. We found that, according to the leaky MVT, mice averaged the past reward with the time constant $\kappa = 0.86 \pm 0.04$, perceived the switching cost as $\langle c_{sw} \rangle = 2.3 \pm 1.3 \mu l$, and exhibited the perceptual noise of $\alpha = 0.33 \pm 0.24$. In the revised paper, we will use these parameters in our deep R-model for quantitative similarity between the data and the model. We further used AIC to confirm that the leaky MVT decision rule explains the data better than the MVT ($\Delta AIC = -101 \pm 62$).

**Neuronal substrates.** We agree that differentiating between the V- and R-models based on TD errors is challenging, nonetheless we argue that consistency between VTA DA and R-learning TD error is an important result in light of the advantages of R-learning. We currently attempt to differentiate between these models using additional manipulations. Note that in R-learning, both expected and unexpected rewards are discounted by the *same* value of an exponentially averaged reward, hence TD error in a converged R-model matches that of the V-model, and is independent from the average reward (lines 234-240). At the same time, theory, model and data all reveal that the exponentially averaged reward varies substantially ($1 - \kappa \approx 0.14$ of reward variation) in contrast with the intuition offered by the reviewer #4.

**Optimality of R-learning and Bayesian inference.** We argue in the discussion that R-learning behavior can be optimal in the real-world settings. Unlike our tasks with repetitive reward patterns, natural environments are dynamic, i.e. the reward richness may change over time. To behave optimally in such settings, the animals have to keep track of the environmental variables such as the average reward rate. This is exactly what happens in the leaky MVT with its short-term averaging of the reward. The optimal timescales for exponential averaging w.r.t the dynamics of an environment have been previously studied in hidden Markov models using Bayesian inference approach (Ref [29]). It has been suggested that animals, evolutionarily exposed to dynamic environments, may have finetuned their decision rules to the average-case environmental dynamics, and use these in fixed experimental settings by extension (Ref [30]). This logic is in line with viewing deep RL networks as meta-learning circuits optimizing the generalized decision rules in their weights (Ref [8]). Although the reviewers are correct to point out that we did not perform Bayesian inference modeling, we connect it to RL via literature on optimality of exponential filtering (cited above / lines 302-320).

**Theoretical implications.** R-learning may be advantageous for computation. In dynamic environments, R-learning – converging to the leaky MVT decision rule – enables agents to adjust to changes in reward contiguity. Such adjusment does not necessitate costly updates of high-dimensional state values. The R-model allows a simple implementation: exponential averaging may be performed with a recurrent neuron. Such threshold update mechanism may not only be computationally efficient ($\mathcal{O}(n)$), but also has potential of outpeforming more complex methods in time-varying stochastic environments. This is akin to (Sutton, 1992) showing that an $\mathcal{O}(n)$ gain tuning beats the $\mathcal{O}(n^2)$ Kalman filter in dymanic environments, because the latter is only optimal if the model of the world is precise. R-learning also broadens the class of models for DA activity, potentially impacting interpretation of a range of experimental findings.

**Relation to foraging and neuroeconomics literature.** Our work builds upon results in the field including Ref [2] mentioned by the reviewer #1; we will further clarify this in the discussion. Two key differences are the comparison of a broader class of models and the the choice of species, allowing us to monitor and manipulate neuronal activity. There are critical differences in the reward schedule: Ref [2] manipulates low/high initial reward values in blocks, whereas we draw random initial rewards intermittently. Our design aims to: 1) reduce autocorrelation in reward sequences to enable analyses as logistic regression, and 2) disentangle the impact of the initial and average reward. Each initial reward is preceeded with the same variety of reward sequences – yet the average leaving thresholds are unique for each initial value. This observation enabled us to pursue the hypothesis of the leaky estimate of average reward. Our approach, summarized in the leaky MVT rule, explains overharvesting (Ref [2]) mentioned by the reviewer #1. Specifically, the exponentially averaged reward drops in parallel with the current reward – after which they meet at a lower threshold than predicted by the MVT. The models used in our work are similar to these in (Ref [2]) with the exception of the definition of state = history of the rewards, which, we argue, allowed our models to generalize predictions over various reward schedules. In a revision we will clarify these issues and also discuss the relation to the previous studies including Ref [2], and the prospect theory / reference point literature (e.g. Mobbs et al, 2018; Constantinople et al, 2019).

[Meta-Review · NeurIPS 2020]

This is a well-written and presented paper proposing a new framework for modeling animal behavior during a foraging task, and should be of interest to the NeurIPS audience. After rebuttal, 3 of the reviewers recommended accept based on it providing a nice link between the behavioral economics and reinforcement learning communities, and its strengths in both theory and empirical results. Therefore, I tentatively recommend accept. That said, during the discussions some concerns were brought up regarding some missing related work. I urge the authors to consider discussing in their final version several related works that R4, and I think are quite relevant: Daw et al, 2002, Neural Networks; Schwighofer & Doya 2003, Neural Networks; Niv et al 2006/2007 (and related), and also some works from motivation modeling literature (that R2 mentions in their review). In the end, I don’t think this is enough to warrant rejection, but it does make this more of a borderline case for me.